# Development of a Generic Domestic Water Security Index, and Its Application in Addis Ababa, Ethiopia

**Yonas T. Assefa [1,2], Mukand S. Babel [1,*], Janez Sušnik [2]**  **and Victor R. Shinde [1]**

1   Water Engineering and Management, School of Engineering and Technology, Asian Institute of Technology, Pathumthani 12120, Thailand; yonasabaye@gmail.com (Y.T.A.); victorshinde@ait.ac.th (V.R.S.)
2   Integrated Water Systems and Governance Department, IHE Delft Institute for Water Education, 2601 DA Delft, The Netherlands; j.susnik@un-ihe.org
*   Correspondence: msbabel@ait.ac.th

**Abstract:** Water security is a global concern because of the growing impact of human activities and climate change on water resources. Studies had been performed at global, country, and city level to assess the water security issues. However, assessment of water security at a domestic scale is lacking. This paper develops a new domestic water security assessment framework accounting for water supply, sanitation, and hygiene through twelve indicators. Water supply, sanitation, and hygiene are central to key water-related sustainable development goals. The framework is subsequently applied to the city of Addis Ababa, Ethiopia. From the domestic water security assessment of Addis Ababa, the water supply dimension was found to be of good level, whereas the sanitation and hygiene dimensions were of poor and fair level, respectively, indicating both a challenge and an opportunity for development. Because the analysis is spatially explicit at the city-branch level (in Addis), variation in domestic water security performance across Addis Ababa can be assessed, allowing efficient targeting of scant resources (financial, technical, personnel). Analysis further shows that a lack of institutional capacity within the utility, existing infrastructure leading to 'lock-in' and hindering maintenance and upgrade initiatives, and an unreliable power supply are the main issues leading to poor domestic water security in the study city. These areas should be tackled to improve the current situation and mitigate future problems. The developed framework is generic enough to be applied to other urban and peri-urban areas, yet provides planners and policy makers with specific information on domestic water security considering water supply, sanitation and hygiene, and accounting for within-city variability. This work could therefore have practical applicability for water service providers.

**Keywords:** developing cities; domestic water security; Ethiopia; hygiene; sanitation; urban water security; water supply

---

## 1. Introduction

"Water security is the capacity of a population to safeguard sustainable access to adequate quantities of acceptable quality water for sustaining livelihoods, human well-being, and socio-economic development, for ensuring protection against water-borne pollution and water-related disasters, and for preserving ecosystems in a climate of peace and political stability" [1]. Securing drinking water supply is one of the fundamental components in urban water security, and delivery of clean water and sanitation is one of the sustainable development goals (SDG6). However, meeting drinking water demands is becoming a big challenge globally, particularly in developing countries, due to an increase in consumption that is driven by urbanization, rapid population growth, economic growth, and change in local climate [2].

Currently, about 1 in 2 people is an urban dweller and by 2030 about 60% of the population are expected to live in cities, partially contributing to an expected water consumption increase of 55% by 2050 [3]. In addition, low-income countries are expected to be the most affected by climate change and the impact on securing water resources [4]. Sub Saharan and eastern Africa are particularly exposed to climate change impacts [5]. Ethiopia is one of the Horn of Africa countries which is currently most exposed to droughts.

It follows that rapidly expanding urban areas in lower income countries face considerable water supply and sanitation challenges in the near future. Addis Ababa, Ethiopia, is no different. The city is home to more than 25% of the urban population of Ethiopia and is one of the fastest growing cities in Africa [6]. Ndaruzaniye [5] reports that investment in the water supply and sanitation coverage has not followed population growth. As a result, the water supply service only covers 55% of the city area, half of the population is served for less than 12 h per day, and a quarter of the population have no formal service at all [5]. More recent studies show that the city still suffers similar water supply problems [7,8].

Although studies have been performed to identify the major factors affecting water security [9,10], a fundamental challenge is developing a general method to measure water security that also yields city-specific information. One of the main reasons making it difficult is that water security is broadly defined and is a combination of various aspects which vary at different scales. For example, Vörösmarty et al. [11] assessed water security on a global scale, considering human water security and biodiversity. On the other hand, the national scale is also commonly used. A common framework for water security assessment at the country level is a framework with five key dimensions, chosen for their simplicity, developed by the Asian Water Development Outlook (AWDO) [12,13]. It has been applied in 49 ADB member countries in Asia and the Pacific.

A recent study in assessing water security at city level was conducted by Babel et al. [14]. The research establishes a framework with five components of water security, and was applied in Bangkok, Thailand. It is shown that the indicators are good in representing water security at the city level in which they considered domestic water security, water productivity, security against water-related disasters, environmental water security, and governance and management aspects. The proportion of piped water supply, water consumption, and proportion of safe drinking water were considered to evaluate the domestic water security. A similar city-level study, but lacking in sub-city detail, was carried out by Jensen and Wu [15].

In most water security assessment studies, domestic water security is studied as one component or dimension. Access to improved water supply and sanitation was taken as an indicator to assess domestic water security [16]. A recent study which solely focuses on the household level domestic water security used the following indicators: piped water coverage, improved sanitation coverage, and hygiene (number of age-standardized disability-adjusted life years (DALYs) per 100,000 people for the incidence of diarrhea) [13].

Measuring the water security at the global or national level will only give the general picture of the situation from a countrywide standpoint, not from a local perspective [17]. Accordingly, assessing the specific water supply and sanitation components at the local level will be helpful for considering detailed challenges and opportunities for the domestic water security situation. According to Babel et al. [14], domestic water security is one of the water security components which needs to be assesssed at the local level.

In most of the assessments, only a few indicators were considered to evaluate the domestic water security of a city. However, those indicators give a limited picture of the challenge on water security. Thus, improved indicators are required to investigate the water security situation at the domestic level. The biggest challenge in assessing domestic water security is the lack of a well-organized framework. This paper develops a domestic water security framework to include water supply, sanitation, and hygiene to determine domestic-level water security, and applies it to Addis Ababa to assess the current

domestic water security situation. Because the developed framework is generic, it can be readily applied to other cities globally while retaining locally relevant information.

## 2. Development of a Domestic Water Security Framework

### 2.1. Definition of Domestic Water Security

Domestic water security is the ability of a population to safeguard sustainable access to adequate quantities of and acceptable quality water for the basic household needs of drinking, sanitation, and hygiene. Although it can be argued that water security is a much broader issue incorporating aspects such as crises and emergency planning and response, addressing vandalism and terrorist acts, development of warning and monitoring systems, ecosystem preservation, and mitigation against water-related hazards, in this paper, the focus is very much on the water security at a household level. It is, therefore, centered on the water supply, sanitation, and hygiene dimensions of domestic water security.

### 2.2. Structure of the Framework

The developed framework has three levels of information for water security assessment: dimensions, indicators, and variables. Dimensions are the various broad components of domestic water security. Numerous indicators are used to represent the dimensions of domestic water security. Variables are then identified to quantify the indicators (Table 1).

**Table 1.** Dimensions, indicators, and definition of variables for this study.

| Dimension, $i$ | Indicator, $j$ | Variable, $k$ | What Is Assessed? | Measurement Unit |
|---|---|---|---|---|
| Water Supply | Availability | The available volume of water resource for drinking purpose | Is the available water sufficient to fulfill the required water demand for the household purpose? | $m^3\,cap^{-1}\,year^{-1}$ |
| | Accessibility | Improved water supply (treated and piped water supply) Water supply service duration | Is the accessibility of water supply service good enough in terms of access to piped water supply and service duration? | % people with access to network Hours of service |
| | Quantity | Adequate water for domestic consumption (per capita consumption) | Is the supplied water adequate enough to satisfy the required water consumption? | $L\,cap^{-1}\,day^{-1}$ |
| | Quality | Acceptable water quality for human health | How well is the supplied water quality for the health of human beings? | $pH$, $mg\,L^{-1}$, NTU |
| | Affordability | Water tariff | Is the cost of water affordable to all types of community? | $US\$\,m^{-3}$ |
| | Water management efficiency | Water loss (NRW) | Is the distributed water reaches efficiently to the consumers? | % NRW in the network |
| Sanitation | Accessibility | Improved sanitation system in terms of customers who use sewerage system | Is the sanitation service good enough in terms of access to improved sanitation coverage? | % people with access to network |
| | Quantity | The amount of wastewater generated | Are the available treatment plants adequate enough to treat the produced wastewater? | $m^3$, $m^3\,cap^{-1}\,year^{-1}$ |
| | Quality | Quality of the effluent | Are the effluent water quality parameters fulfilling the required standards? | $pH$, $mg\,l^{-1}$, NTU |
| | Affordability | Affordability of domestic wastewater collection (tariff) | Is the cost for wastewater transportation affordable to all the communities? | $US\$\,m^{-3}$ |
| Hygiene | Adequacy of water for hygiene | Water availability for hygiene | Is the supplied water adequate to fulfil the required amount for hygiene? | $m^3\,cap^{-1}\,year^{-1}$ |
| | | Awareness | Is there diarrheal problem due to lack of water and poor toilet facilities (lack of hygienic toilet)? | Toilets per number of people |
| | | | Is the population aware of about hygiene? | Questionnaire responses |

The dimensions of domestic water security were established based on the Sustainable Development Goals of clean water and sanitation (SDG6). The dimensions for this study are: (i) water supply; (ii) sanitation; and (iii) hygiene. These three dimensions are represented by numerous indicators, which are measured via the variables (Table 1). More than one variable can be used to assess an indicator. Many indicators combine to represent dimensions. Table 1 details the dimensions,

the associated indicators, and the variables related to the indicators. It also provides definition of each of them. Figure 1 shows how the variables, indicators, and dimensions are related to form the overall domestic water security index. The rest of this section provides more details about each indicator and how it is calculated or assessed in this work.

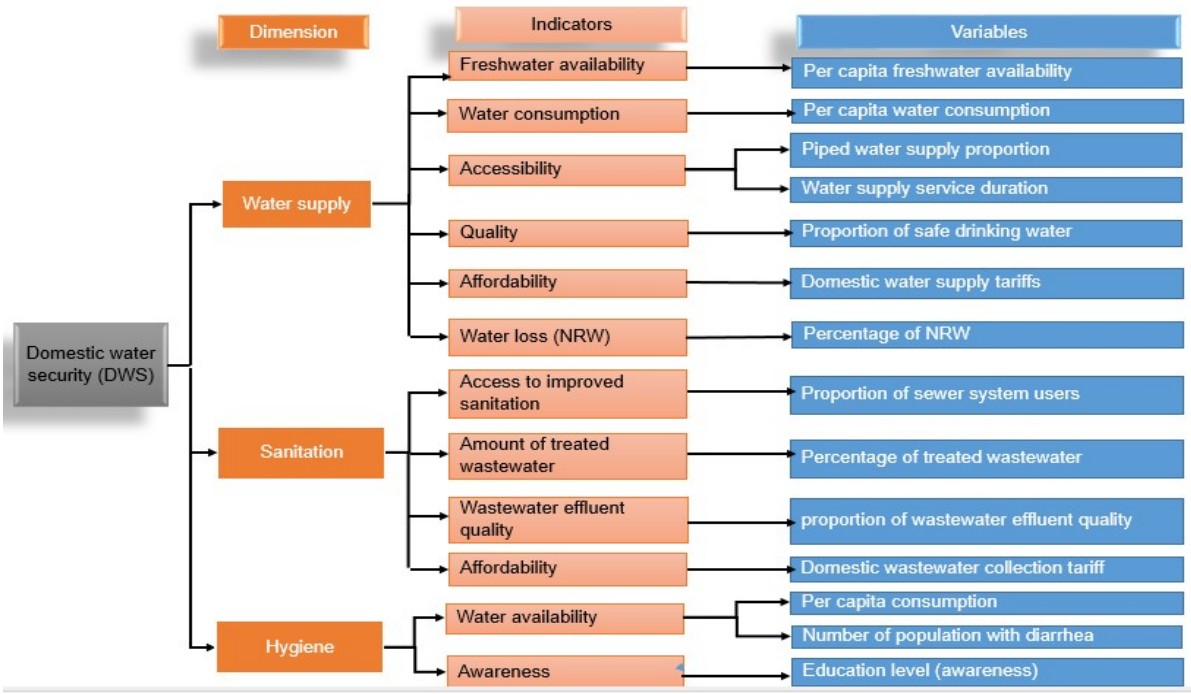

**Figure 1.** The developed domestic water security index framework.

*2.3. Water Supply Dimension*

2.3.1. Per Capita Freshwater Availability

The available water resources should be adequate for different domestic and non-domestic uses. Many studies use the Falkenmark et al. [18] water stress index to measure water scarcity. According to this index, water availability <1000 m$^3$ cap$^{-1}$ year$^{-1}$ indicates water scarcity, while values <500 m$^3$ cap$^{-1}$ year$^{-1}$ represent absolute water scarcity. The per capita water availability can be obtained from:

$$\text{Per capita water availability } = \frac{\text{Water resources}}{\text{Total population}} \tag{1}$$

where per capita water availability has units of m$^3$ cap$^{-1}$ year$^{-1}$, water resources is in m$^3$ year$^{-1}$, and the population is number of people.

2.3.2. Per Capita Water Consumption

This variable is used to determine the current state of water consumption in routine household activities. According to Howard and Bartram [19], with a consumption of 100 L cap$^{-1}$ day$^{-1}$ at least all basic human needs are met, including hygiene. The consumption can be assessed as:

$$\text{Per capita water consumption } = \frac{\text{Water consumed}}{\text{Total population}} \tag{2}$$

where water consumed is measured in L cap$^{-1}$ year$^{-1}$.

### 2.3.3. The Proportion of Piped-Water Supply Users with Respect to the Total Population

According to Hsu et al. [20], improved water supply means that a facility or delivery point protects water from external contamination—particularly fecal contamination. This includes piped water to consumers. Therefore, the proportion of the population having piped water supply must be calculated in order to assess the status of improved water supply coverage of the city.

$$\text{Proportion of piped water supply} = \frac{\text{No. of piped water supply users}}{\text{Total population}} \times 100 \tag{3}$$

where the proportion is expressed in % of total population.

### 2.3.4. Water Supply Service Duration per Day

This factor is included in order to determine the duration of accessing improved water supply and continuity of the water supply service of the city, i.e., reliability of water supply. It is assumed that in order to say the city is excellently secured in terms of water supply, customers should have access to water for 24 h per day.

### 2.3.5. The Proportion of Safe Drinking Water Supply Based on Drinking Water Quality Standards

Since the quality of water directly impacts human health, it is fundamental to assess the quality of supplied water. According to WHO [21], the recommended parameters of drinking water quality in community water supplies should include *E. coli*, residual chlorine, pH value, and turbidity. Therefore, those parameters are assessed in order to check the status of the quality of water at each of the treatment plants in the study area. The WHO [21] drinking water quality standards state that *E. coli* must be absent, residual chlorine must be greater than or equal to $0.2 \text{ mg L}^{-1}$, pH value must be between 6.5 and 8.5, and turbidity must be equal or less than 5 NTU. Data on other parameters such as nitrate, pesticides, and heavy metals within the drinking water supply were not available in this study, but could be added in other settings if desired.

### 2.3.6. Affordability of Domestic Water Supply Tariff

Affordability means that the water supply should consider low-income people so that all of the community can access an improved water supply. A tariff of about $0.40 \text{ US\$ m}^{-3}$ is taken as a benchmark for this study, following the study of Banerjee and Morella (2011), who propose this as an affordable tariff in developing countries. While $0.40 \text{ US\$ m}^{-3}$ is suitable for this particular case, for other applications a relative benchmark such as the amount of income (as a proportion) spent on water services may be more appropriate. In this case, such data were not available, so the $0.40 \text{ US\$ m}^{-3}$ value is taken. It is also important to note that this number may not be applicable to other locations, therefore site-specific information is required. Also for this study, time and budget prohibited in-depth investigation of a site-specific tariff (or related metric).

### 2.3.7. Percentage of Non-Revenue Water (NRW)

Water loss in the water supply system is a major issue in many developing countries. High water loss leads to shortage of water to the customers and increased costs for the operator. Thus, the water loss of the city should be assessed to determine the water security status of the city.

### *2.4. Sanitation Dimension*

### 2.4.1. The Proportion of Customers Connected to the Sewer System

Access to sanitation was measured by considering the population that has access to an improved sanitation system. According to Hsu et al. [20], improved sanitation is defined as access to a connection to a sewer system, connection to a septic system, ventilated pit latrine, and pour-flush latrine. In general,

if the system hygienically separates human excreta from contact and is not public, it is considered improved. For this study, the proportion of sewer line system users was considered in order to evaluate improved sanitation access. Coverage of improved sanitation in terms of sewer line system is calculated based on the number of customers connected to the sewerage system to the total population of the area.

$$\text{Proportion of sewer line users} = \frac{\text{No. of sewer line users}}{\text{Total population}} \times 100 \qquad (4)$$

2.4.2. Percentage of Treated Wastewater

About 80% of the freshwater supplied to the customers is assumed to be returned to the drainage network as wastewater. This wastewater should be treated in order to protect human health and the environment. Thus, determining the amount of treated wastewater will lead us to see the water security status of the city. A city with 100% wastewater treatment capacity will be the most secured city from this perspective.

2.4.3. Proportion of Wastewater Effluent Quality Based on Wastewater Discharge Quality Standards

The quality of the treated wastewater should be within the recommended range in order to keep human health and to protect the environment. Assessing the quality of effluent will be helpful in determining the water security status of the study area.

2.4.4. Affordability of Domestic Wastewater Collection Tariff

Collection of wastewater through a sewerage system should be affordable to all. This will help in improving the health status of people. Therefore, it is vital to assess the affordability of the wastewater disposal tariff of the study area. This can be assessed based on a questionnaire survey, although time and budget constraints meant such a survey was not possible here.

*2.5. Hygiene Dimension*

2.5.1. Water Availability for Hygiene (per Capita Water Consumption)

Availability of water is fundamental to human health. According to Howard and Bartram [19], a consumption of 100 L cap$^{-1}$ day$^{-1}$ covers all basic human needs, including hygiene. This metric is measured in the same way as for 2.3.2.

2.5.2. Percentage of Population with Diarrhea

Having a proper toilet system and hand wash facilities with adequate water will improve the hygiene status of a community. Diarrhea is an indicator of a non-hygienic situation. This is a common disease in developing countries. Thus, the percentage of people with diarrhea is a good representative to reflect the water security of a place. This can be obtained through household surveys or from secondary data from health centers and hospitals.

2.5.3. Education Level

Awareness is a basic requirement of the hygienic condition. In this study, it is assumed that more educated persons will have good hygienic conditions compared to less educated persons. Therefore, education level is used as a proxy for hygiene conditions of people.

*2.6. Data*

All the required data to assess the current domestic water security status of Addis Ababa is secondary data which were obtained from different Ethiopian governmental offices and national statistical datasets. In depth questionnaires were not possible due to time and budget constraints.

Most of the data were obtained from the head office of Addis Ababa Water Supply and Sewerage Authority (AAWSA) during a short field visit in late 2017. The data are collected based on the eight utility's branches of AAWSA for a period of five years from 2013 to 2017, and consist of:

Capacity of the water supply sources;
Number of registered customers who use the piped water supply;
The water supply service duration of each branch (hours/day);
Bill data of each branch to determine the amount of water actually consumed;
Water quality report of treatment plants;
Tariff for domestic water supply;
Capacity of wastewater treatment plants;
Number of sewer system users.

*2.7. Representation and Interpretation of the Domestic Water Security Index*

Since the variables and indicators have different units, it is desirable to develop a numerical value to represent the indicators on a common scale (i.e., normalization). Also, this scale should have an interpretation so that it will be easy and helpful to communicate with different stakeholders. Table 2 details the representation of each variable described above, classifying the obtained values into a 1–5 scale. Table 3 shows the interpretation of the different domestic water security index scores on the 1–5 scale, where <1.5 is poor, and >4.5 is excellent.

**Table 2.** Representation of variable scores in relation to the 1–5 scale adopted for this study.

| Dimension | No. | Variable | Unit | Representations (Normalization) | | | | | Reference |
|---|---|---|---|---|---|---|---|---|---|
| | | | | 1 | 2 | 3 | 4 | 5 | |
| Water Supply | 1 | Per capita fresh water availability | $m^3\ cap^{-1}\ year^{-1}$ | <500 | 500–800 | 800–1000 | 1000–1700 | >1700 | [18] |
| | 2 | Per capita water consumption | $L\ cap^{-1}\ day^{-1}$ | ≤20 | 21–50 | 51–90 | 91–100 | ≥101 | [19] |
| | 3 | The proportion of piped water supply users to the total population | % | 0-60 | 61–70 | 71–80 | 81–90 | 91–100 | [12] |
| | 4 | Water supply service duration per day | h | <8 | 8–16 | 17–20 | 21–23 | 24 | This study* |
| | 5 | The proportion of safe drinking water supply based on drinking water quality standards | % | 0-60 | 61–70 | 71–80 | 81–90 | 91–100 | [21] |
| | 6 | Affordability of domestic water supply tariff | US$ m$^{-3}$ | >1 | 1 | 0.75 | 0.5 | <0.4 | [22] |
| | 7 | Percentage of NRW | % | ≥25 | 25–20 | 20–15 | 15–10 | 10–0 | [23] |
| Sanitation | 8 | The proportion of customers who uses sewer line system | % | 0-60 | 61–70 | 71–80 | 81–90 | 91–100 | [12] |
| | 9 | Percentage of treated wastewater | % | 0-60 | 61–70 | 71–80 | 81–90 | 91–100 | Assumed to be the same as sewer system coverage |
| Hygiene | 10 | The proportion of wastewater effluent quality based on wastewater discharge quality standards | % | 0-60 | 61–70 | 71–80 | 81–90 | 91–100 | [21] |
| | 11 | Affordability of domestic wastewater collection tariff | - | - | - | - | - | - | Assumed to be checked by questionnaire |
| | 12 | Water availability for hygiene (per capita consumption) | $l\ cap^{-1}\ day^{-1}$ | ≤20 | 21–50 | 51–90 | 91–100 | ≥101 | [19] |
| | 13 | Number of population with diarrhea | DALYs | ≥1000 | 800–500 | 500–100 | 100–30 | ≤30 | Or Household survey and health centers [13] |
| | 14 | Percentage of educated people (awareness) | % | - | - | - | - | - | Assumed to be checked by questionnaire |

Note: (DALYs) is number of age-standardized disability-adjusted life years per 100,000 people for the incidence of diarrhea; *Developed in this study by assuming the minimum time to collect water as 8 h.

**Table 3.** Interpretation of the domestic water security index scores.

| Index | Level of Water Security | Interpretation |
|---|---|---|
| (<1.5) | Poor | The city is incapable of meeting the basic water requirements for its citizens. Water is used indiscriminately without proper planning and management that is a serious point of concern for all dimensions of water security. |
| (1.5–2.5) | Fair | The actions required to ensure water security are evident. However, there are still major gaps and serious concerns in regards to almost all dimensions of water security. |
| (2.5–3.5) | Good | The city has a fairly satisfactory system and environment for facilitating water security. However, some dimensions of water security are still a cause of concern. |
| (3.5–4.5) | Very good | The city is well-placed with most of the dimensions of water security. Their security against the dimensions may not be at par with the others, but the overall situation is still nonetheless very goodly satisfactory. |
| (>4.5) | Excellent | The city is an ideal example of a water-secure society. It shows exemplary levels of security against every dimension of water security. |

*2.8. Measuring Domestic Water Security*

In order to get the index of overall water security, the values of variables, indicators, and dimensions must be aggregated [24]. For this study, equal weights to all variables, indicators, and dimensions are applied because addressing or weighing elements that are having unrelated issues with different units is complicated and may lead to misinterpretation [25]. The aggregation equations were adopted from Onsomkrit [24], who carried out water security assessments at the city scale.

The first step in the calculation of the domestic water security index is to give a score for each of the variables (*k*) according to their values as per Table 2. After all the variables (*k*) are assigned with their respective scores, the next step is to calculate the indicators (*j*), dimensions (*i*), and overall domestic water security index (*DWSI*). This is detailed here.

The value of an indicator (*I<sub>ij</sub>*) is calculated with respect to scores given to variables as below:

$$I_{ij} = \sum_{j=1}^{n} \sum_{k=1}^{m} w_{ijk} \times V_{ijk} \tag{5}$$

$I_{ij}$: value of indicator *j* for dimension *i*; *n*: no. of indicators in dimension *i*; *m*: no. of variables in indicator *j*; $w_{ijk}$: weight given to variable *k* of indicator *j* of dimension *i*; $V_{ijk}$: score of variable *k* of indicator *j* of dimension *i*.

The total of weights given to the variables equals 1.

The value of a dimension (*D<sub>i</sub>*) is calculated with respect to values of indicators:

$$D_i = \sum_{j=1}^{n} w_{ij} \times I_{ij} \tag{6}$$

where $D_i$ is value of dimension *I*, $w_{ij}$ is weight given to indicator *j* of dimension *i*. The total of weights given to indicators equals 1.

The domestic water security index (*DWSI*) is calculated with respect to values of dimensions:

$$DWSI = \sum_{i=1}^{p} w_i \times D_i \tag{7}$$

where *DWSI* is domestic water security index, *p* is no. of dimensions, $w_i$ is weight given to dimension *i*. The total of weights given to dimensions again equals 1.

## 3. Application of the Domestic Water Security Index

The developed domestic water security framework was applied to the city of Addis Ababa, Ethiopia (Figure 2). Addis Ababa is located in the central plateau of Ethiopia in the central part of the country. The living standard of the population is the same as many developing cities, and contains

both modern areas (well designed and constructed houses) and slum areas. The city has a total land area of about 520 km$^2$ and is sub-divided into ten sub-city (administrative regions) areas. From the census data of 2007 the population of the city was 2,739,551 with a growth rate of 2.10%.

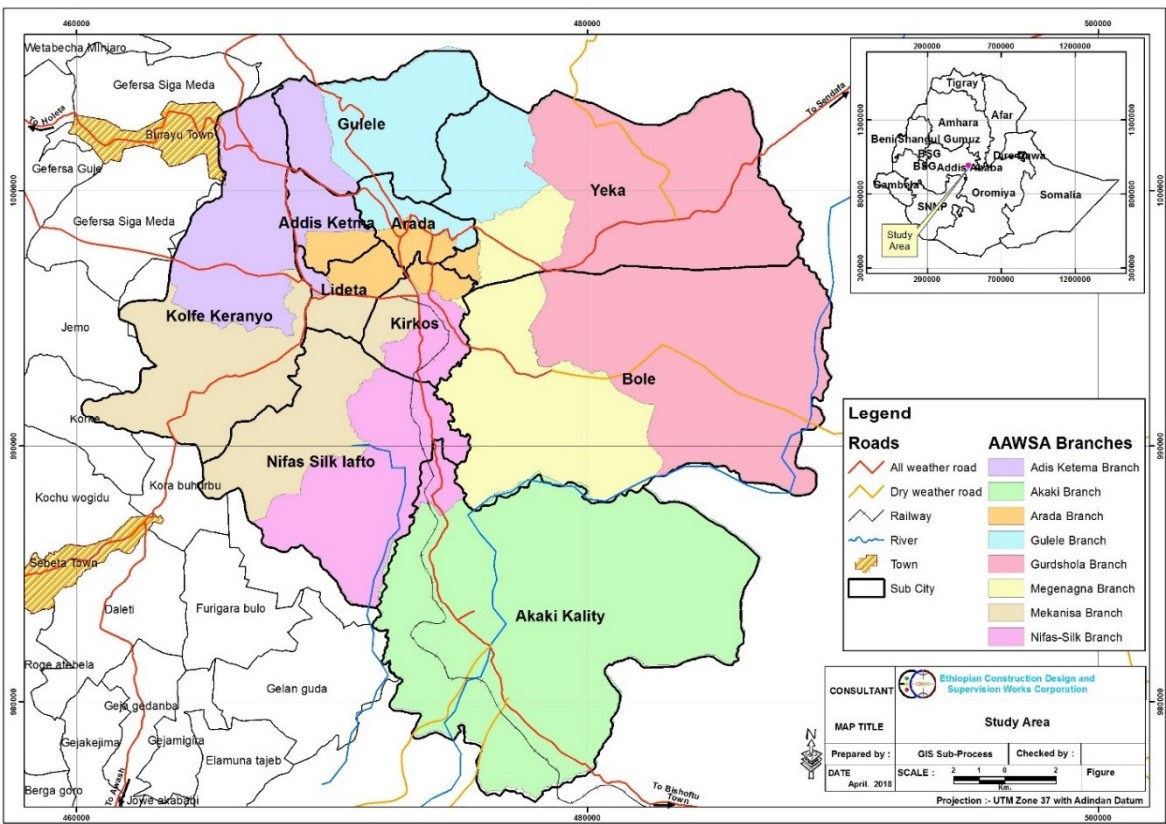

**Figure 2.** Map of Addis Ababa, Ethiopia.

The water supply and sewerage system of the city is governed by Addis Ababa Water Supply and Sewerage Authority (AAWSA). The utility operates the water supply and sewerage services in the city by dividing the city into eight branch areas (Figure 2). The city uses water from both surface and groundwater sources. Currently, it is difficult to differentiate which part of the city gets water from which source (the system is mixed). However, it is possible to say that generally the eastern and northern parts of the city uses more surface water resource (from Gefersa dam), the western and middle part of the city receives both surface and groundwater (Legedadi dam and Legedadi groundwater), and the southern part gets water mainly from groundwater (Akaki welfield).

Due to rapid urbanization, the lifestyle of city dwellers is changing, and many new condominiums are under development. Since these condominiums include modern water and sanitary facilities, they consume more water, increasing demand on the city water supply. In addition, the city is undertaking many expansion and construction projects (e.g., industrial, offices, and international hotels). Due to this, there is a huge migration from the rural part of the country [26], further increasing the water consumption and creating water supply and sanitation service challenges.

## 4. Application and Results

The developed domestic water security index framework (Figure 1) was applied for Addis Ababa in order to assess the existing domestic water security status of the city. From the initial 14 variables as proposed in the framework, nine were used to assess the domestic water security status of the study area based on the availability of data. For the other five, local data were unavailable. The assessment was done at the supply branch level of AAWSA (Figure 2) using data for the period

2013–2017. Local data were not available for the following five variables: per capita freshwater availability; proportion of wastewater effluent quality; affordability of domestic wastewater collection tariff; number of population with diarrhea; education level (awareness). The other nine were assessed according to the framework above. It would have been preferable to have data for all 14 variables, however the framework was developed before the application to Addis, and local data availability was not known prior to collection. Therefore, in this application, five of the 14 variables are not quantified. For some, such as the rate of diarrhea, a recent study shows a very high prevalence throughout Addis [27]. However, the study by Adane et al. [27] is restricted to children aged 0–50 months. As young children are more susceptible to such disease, it is likely that the numbers from the study are overestimates with respect to the entire population. As a consequence, this indicator value would be worse than is actually the case. It was chosen therefore not to use this study in the present work. In other cities, data may be available for all 14 variables, or for fewer than the nine assessed here. Because the framework is described here, future studies could conduct a "data screening" prior to application in order to assess data availability. As long as most variables can be assessed, and that all indicators and dimensions can be quantified to some extent, then a relatively detailed impression of domestic water security in a city can be gained using this framework. By not prescribing that all 14 variables must be quantified, the transferability and flexibility of this framework is increased.

### 4.1. Water Supply Dimension

The water supply dimension was assessed on the following variables: proportion of population with piped water supply; water supply service duration; per capita water consumption; percentage of NRW; conforming to water quality standards; affordability of domestic water supply tariff.

### 4.1.1. The Proportion of Piped Water Supply Users to the Total Population

The proportion of the population with access to the piped water supply system in each branch in 2017 is shown in Figure 3a. Figure 4 shows the change in water supply coverage in terms of the piped water supply of each branch for the last five years (2013–2017).

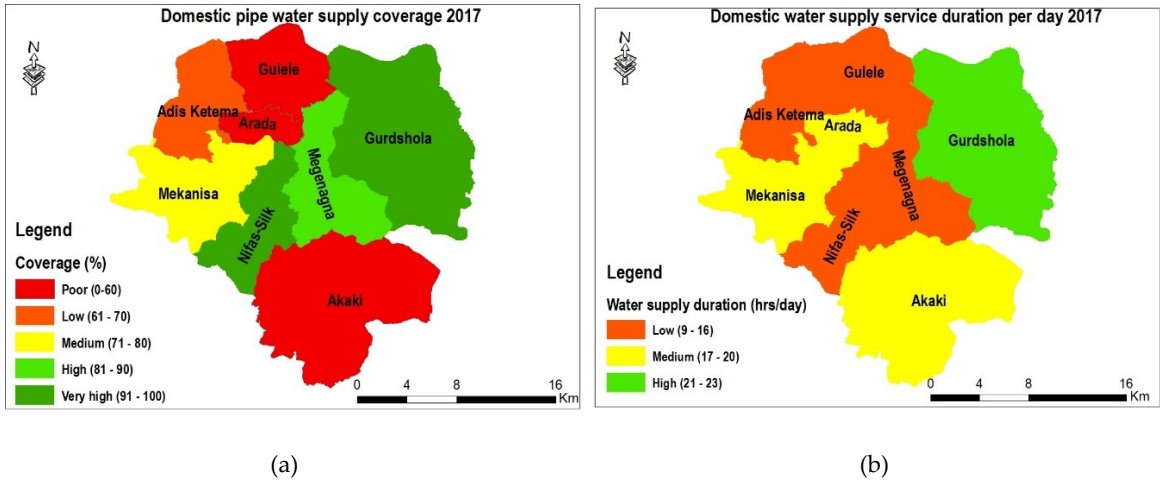

(a)　　　　　　　　　　　　　　　　　　　　　　　　　　(b)

**Figure 3.** *Cont.*

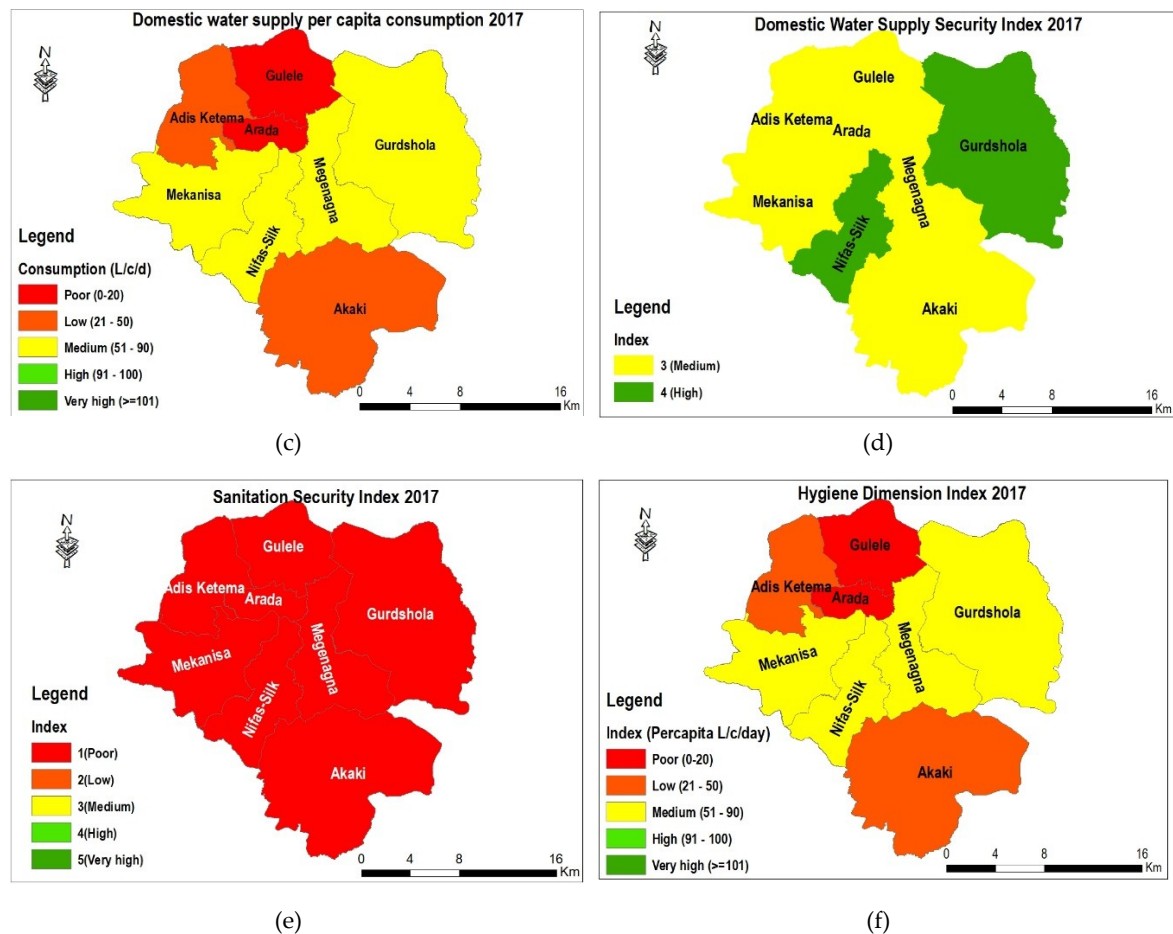

**Figure 3.** Maps showing: (**a**) domestic piped water supply coverage (red: 0–60%; orange: 61–70%; yellow: 70–80%; light green: 81–90%; dark green: 91–100%); (**b**) domestic water supply duration per day (orange: 9–16 h; yellow: 17–20 h; green: 21–24 h); (**c**) per capita water consumption (red: 0–20 L cap$^{-1}$ day$^{-1}$; orange: 21–50 L cap$^{-1}$ day$^{-1}$; yellow: 51–90 L cap$^{-1}$ day$^{-1}$); (**d**) the aggregated water supply index (yellow: medium; green: high); (**e**) the aggregated sanitation index (red: poor); and (**f**) the hygiene index (red: poor; orange: low; yellow: medium) for each city branch in Addis Ababa in 2017. See Table 2 for an interpretation of the index values and Table 3 for quantification of each index value with respect to the corresponding variable.

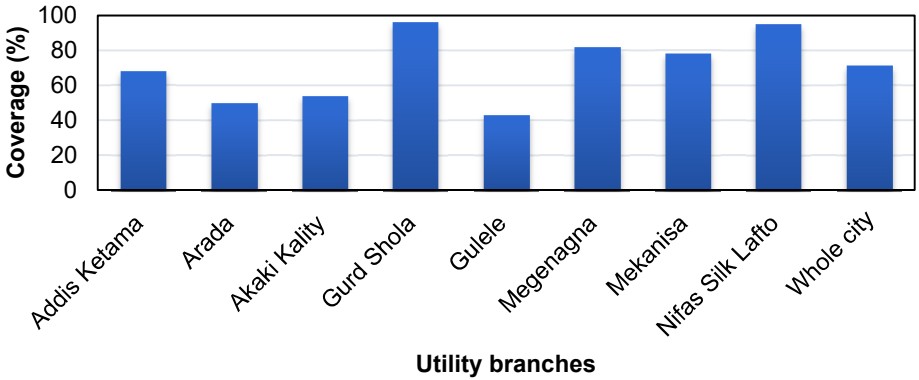

**Figure 4.** Proportion of piped water supply users of each branch in 2017.

Gurd Shola and Nifas Silk Lafto branches reach 95% piped water supply coverage in 2017, whereas Gulele and Arada branches show steady proportions with less than 50% coverage. Overall, the proportion of improved domestic piped water supply coverage of Addis Ababa in 2017 is about 71%.

### 4.1.2. Water Supply Service Duration per Day

Water supply service duration in each branch was assessed (Figure 3b). Addis Ababa is using an intermittent water supply service system and each branch has its own schedule. The schedule was collected from the branch offices. According to the collected data, all branches receive water for less than 24 h. Currently, customers in Gurd Shola branch get water for about 23 h in a day, while customers in Megenagna and Gulele branches receive water for only 11 h a day. Generally, Addis Ababa residents get water for about 17 h per day.

### 4.1.3. Per Capita Water Consumption

The domestic per capita water consumption of each branch from 2013–2017 is shown in Figure 5, and in Figure 3c. The city-wide domestic per capita consumption was about 40 L cap$^{-1}$ day$^{-1}$ in 2015 [6], very close to that obtained in this study.

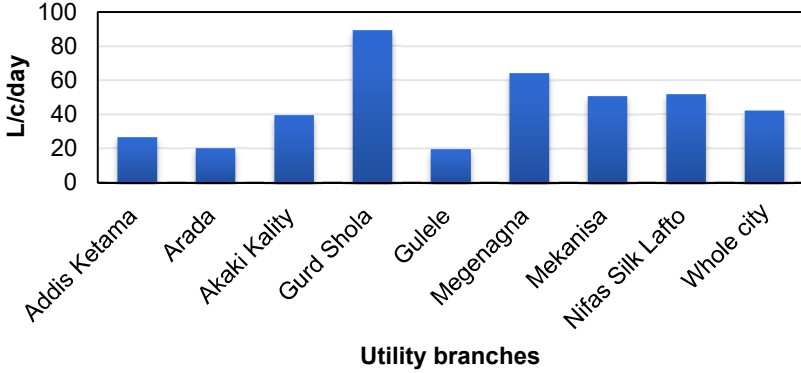

**Figure 5.** Per capita water consumption (L cap$^{-1}$ day$^{-1}$) of each utility branch based on bill data in 2017.

Figure 5 shows that the current per capita consumption in Addis Ababa varies between 18 L cap$^{-1}$ day$^{-1}$ (Gulele branch) and 89.25 18 L cap$^{-1}$ day$^{-1}$ (Gurd Shola branch). Possible reasons for this variation are explained in the discussion section (Table 4). The estimated per capita consumption by this study is less than from the utility plan. It is shown that the actual consumption is about half of that projected by the AAWSA business plan, suggesting demand is not met. It is noted that the per-capita water consumption figures are very low. One explanation is that the city suffers with huge water losses (about 40%). In addition, domestic water supply service duration is <24 h, where many locations only receive water for parts of a day. Therefore actual supply is indeed low. At the same time, locals supplement their supply using alternative sources. Although there are no formal alternative water supply systems, the majority of communities use different storage systems to satisfy their needs (e.g., plastic tanks). In addition, people with higher incomes use bottled water (a rapidly increasing trend in Ethiopia) from shops, mainly for drinking purposes.

### 4.1.4. Percentage of Water Loss (NRW)

Non-revenue water (NRW) data was obtained from AAWSA. NRW of the city for the last five years was almost constant at about 40%. Branch-level data were not available; therefore, the city-wide number was adopted for each branch. According to the utility target, it was planned to reduce NRW to 20% by the year 2020 [28], however, the trend doesn't seem to achieve the planned target.

### 4.1.5. Conforming to Water Quality Standards

The water quality parameters of pH, turbidity, residual chlorine, and *E. coli* of the treatment plants were obtained for 2017 at each month from AAWSA databases. The data obtained were at a monthly resolution. During collection, no further information was provided regarding variability in

the measurements or water quality violations. These parameters were compared to WHO standards and results show that the parameters were found to be within required standards. Thus, the supplied water is of good quality after treatment. The water quality of the city was taken as the same for all branches due to data limitations (data were not available at branch level).

### 4.1.6. Affordability of Domestic Water Supply Tariff

According to Banerjee and Morella [22], tariffs of about US$ 0.40 m$^{-3}$ are considered sufficient to cover operating costs in most developing-country contexts, while US$ 1.00 would cover both operation and maintenance (O&M), and infrastructure costs. Thus, using the current exchange rate (27.07 ETB per 1 US$), O&M costs of 10.83 ETB and a full cost recovery tariff of about 27.07 ETB m$^{-3}$ are affordable for developing countries like Ethiopia. However, the average domestic water tariff of Addis Ababa for the last six years is the same, being about 2.85 ETB m$^{-3}$ (0.11 US$ m$^{-3}$). Therefore, the tariff is affordable in Addis, although a questionnaire with residents would be required in order to properly assess the affordability of water supply. O&M, however, is not covered by this tariff, potentially leading to water supply system degradation.

### 4.1.7. Domestic Water Security Sub-Index for the Water Supply Dimension

By using the above variables and indicators and Equation (6), the domestic water security sub-index of each branch was computed for the water supply dimension (Figures 3d and 6).

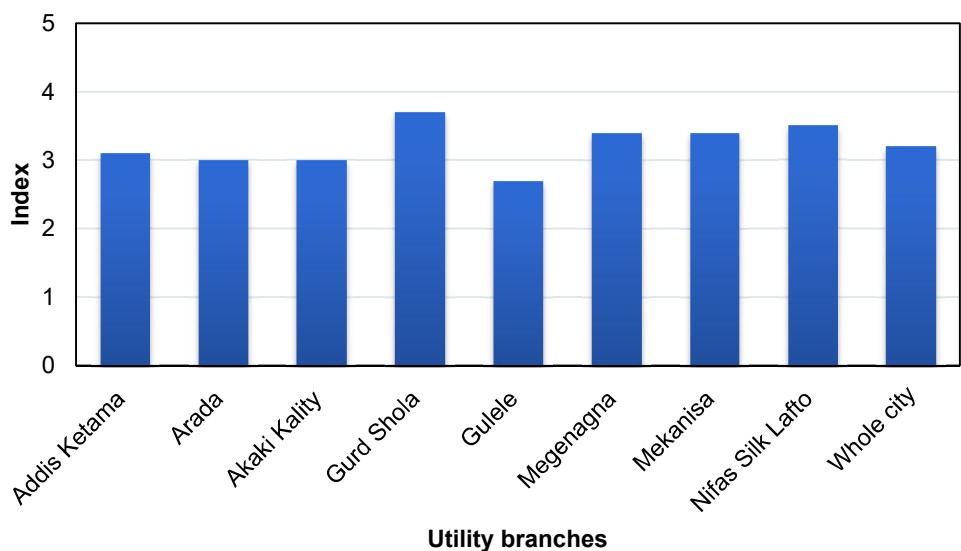

**Figure 6.** Domestic water security sub-index for water supply dimension for each branch in 2017.

From Figure 6, it is seen that the domestic water supply security index of each branch varies between 2.6 and 3.7, which is an indication of good progress towards very good levels of domestic water security level. The Gurd Shola, Megenagna, Nefas Silk Laftos and Mekanisa branches are at good levels of domestic water supply security. The domestic water supply security index of the whole city is 3.20 in 2017. This shows that the water security of the city in terms of water supply is at the "good" level (Table 3). The good water supply security level indicates that the city has a fairly satisfactory system and environment for facilitating water supply. However, some factors of water security are still a cause of concern. From those, the per capita water consumption, challenges in providing access to piped water supply, lower than 24 h uninterrupted water supply, and high non-revenue water loss are the major points of concern.

Even though the overall water supply situation in Addis Ababa shows a good domestic water security level, progress in reaching the whole community with sufficient amount of water (per capita consumption) for 24 hours is still not satisfactory all over the city. From field observation and

discussions with AAWSA head office water supply experts, the major water supply challenges are rapid urbanization, construction, population growth, and migration from rural areas to the city leading to informal settlements. This results in increasing the water demand consumption. In addition, shortage of water supply sources is also another major concern for the utility.

*4.2. Sanitation Dimension*

The sanitation dimension was assessed using two indicators based on the improved sanitation coverage and percentage of treated wastewater variables.

4.2.1. The Proportion of Customers Connected to Sewer System

The proportion of the population connected to the sewer system was determined at each branch. The number of customers was obtained from AAWSA head office for each branch. The city-average improved sanitation coverage is below 12.9% (Figure 7). A small improvement every year in all of the branches is observed. The Megenagna branch shows the highest coverage (with about 35% in 2017), followed by Mekanisa branch (with about 24% in 2017). The minimum sewer coverage is shown in Addis Ketema (with coverage of about 4.8% in 2017).

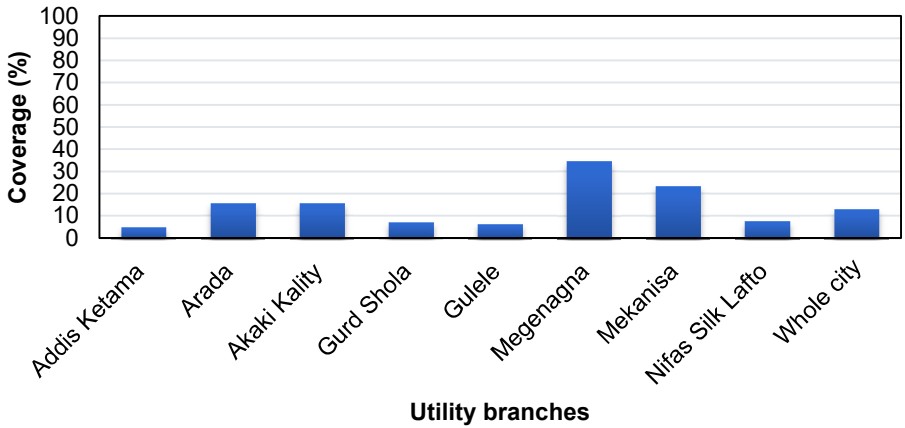

**Figure 7.** Sewer line coverage of each branch in 2017.

4.2.2. The Percentage of Treated Wastewater

The wastewater treatment assessment was performed by comparing the wastewater produced with the wastewater treatment capacity. As mentioned earlier, wastewater production was calculated by assuming 80% of consumed water is returned as wastewater. There is only one wastewater treatment plant in Addis Ababa (Kaliti plant), with a capacity of about 7600 $m^3$ $day^{-1}$ or 228,000 $m^3$ $month^{-1}$. The amount of wastewater treated is very small (<10%). While the wastewater produced has increased, the treatment plant capacity has remained the same for the past five years. Based on this, the percentage of the wastewater treated is obtained for the whole city and adopted the same as to all branches. The average treated wastewater percentage is below 10% (7.03% in 2017). Thus, the value for this variable is 1.

4.2.3. The Overall Sanitation Index

After determining the score of each variable in the sanitation dimension, the next step was to assess the domestic water surety sub-index for the sanitation dimension (Figure 3e). It is shown that the sanitation progress of the city remains steady with the index value of 1, which is poor. The poor sanitation sub-index indicates that the city is incapable of meeting the basic sanitation requirements for its citizens.

### 4.3. Hygiene Dimension

This dimension was evaluated based on only one variable: water available for hygiene.

The Available Water for Hygiene

The available water for basic hygiene is assumed to depend on the water consumption. Areas consuming more water are considered as more hygienic than those with water shortages or where water use is lower. Figure 3f shows the hygiene index of Addis Ababa at each branch for the year 2017. This index has not changed value over the last five years.

### 4.4. Overall Domestic Water Security Assessment

Figure 8 shows the results of water supply, sanitation, and hygiene dimension sub-indices for different city branches in Addis Ababa. The city generally performs better on the water supply dimension than on the sanitation and hygiene dimensions in all city branches.

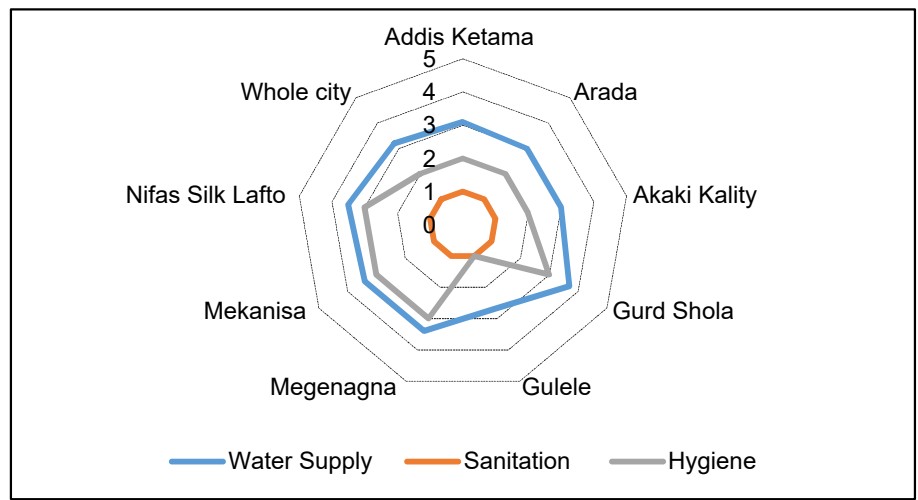

**Figure 8.** Domestic water security index of each of the dimensions for branch in 2017.

The overall domestic water security index is computed from the domestic water security index results of all three dimensions (water supply, sanitation, and hygiene, Figure 9). Although the water security in terms of domestic water supply is good, the overall domestic water security is below the good level due largely to the poor performance of the sanitation and hygiene dimensions. It is seen (Figure 9) that all of the branches remain below the "good" domestic water security level. The overall domestic water security index (*DWSI*) of Addis Ababa is almost constant for the last five years. As per the definition of the class for water security (Table 3), fair domestic water security level means that the essential elements to ensure water security are evident. However, there are major gaps and serious concerns in regard to almost all of the selected variables of the domestic water security. Results indicate that Addis Ababa has to address some fundamental water supply and sanitation challenges, especially shortage of water supply (sources), water loss (high NRW), providing safe sanitation facilities, and issues surrounding financial means in order to maintain and expand water supply and sanitation infrastructure (tariff related).

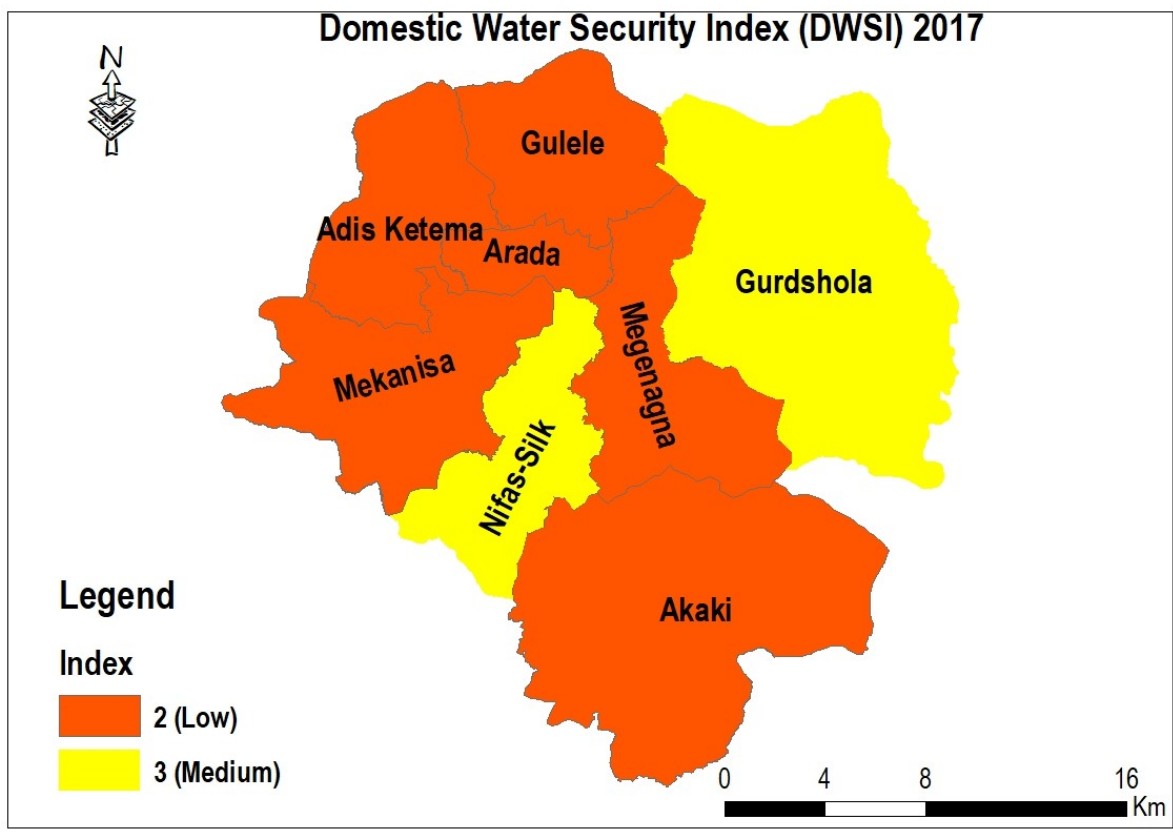

**Figure 9.** Domestic water security index for Addis Ababa in 2017. Orange = low; yellow = medium. See Table 3 for definitions.

## 5. Discussion

Based on the above results and understanding of the study area, Table 4 summarizes the major challenges for improving the domestic water security for each of the eight branches.

This study is the first to develop and apply a framework for the assessment of domestic water security accounting for water supply, sanitation, and hygiene. It is more detailed and comprehensive than national or regional frameworks of water security, and is directly applicable to domestic urban water supply. In addition, the framework is generic, and can be applied with some adaptation, if necessary, to other urban and maybe peri-urban areas where data are available.

Regarding Addis Ababa, Table 4 indicates that issues relating to (institutional) capacity dominate potential reasons for overall poor system security. These issues include a lack of finance available to support adequate maintenance, improvement, or expansion of existing systems, and a lack of personnel to properly register all customers on service networks and to provide maintenance. Increased financial capacity would allow utilities to implement infrastructure repairs, upgrades, and expansion, which in turn could address the high NRW figures in Addis Ababa and improve city-wide customer connection numbers both to supply and sanitation services. However, boosting capacity (financial, personnel, technical), while potentially offering multiple benefits regarding domestic water security, is a considerable challenge throughout developing nations. There is a need to make finance more easily available, but also to develop the suitable technical capacity within the nations in order to be able to deal with existing and upcoming challenges related to rapid urban expansion and water service pressures.

**Table 4.** Discussion on possible reasons for the domestic water security (DWS) values at each branch for 2017.

| Branch | Variable | Value | DWS Score | DWS Level | Major Challenges |
|---|---|---|---|---|---|
| Addis Ketama | Proportion of piped water supply coverage (%) | 68 | 2 | Fair | Financial and other institutional problems<br>Problems with registration of number of customers |
| | Water supply service duration (h day$^{-1}$) | 16.30 | 3 | Good | Topography (Highly elevated area with elevation varying from 2270 to 2895 m above mean sea level)<br>Power supply shortage (to pump the water) |
| | Per capita water consumption (L cap$^{-1}$ day$^{-1}$) | 26.43 | 2 | Fair | Population density is high in this area (more than 587,000).<br>High water loss (NRW) |
| | The proportion of customers connected to the sewer system (%) | 4.78 | 1 | Poor | Maybe difficult to construct sewer system due to the existence of pre-constructed infrastructures in most of the areas. |
| Arada | Proportion of piped water supply coverage (%) | 50 | 1 | Poor | Slum dwellers.<br>Household size might be high in this area due to dense population (number of customers small but many consumers)<br>Financial and other institutional problems<br>Problem with registration of number of consumers |
| | Water supply service duration (h day$^{-1}$) | 18.90 | 3 | Good | Power supply shortage (to pump the water) |
| | Per capita water consumption (L cap$^{-1}$ day$^{-1}$) | 20.15 | 2 | Fair | Densely populated area (the population is high and the area is very small compared to other branches)<br>High economic activity (commercial center of the city)<br>The domestic and non-domestic consumption is almost equal<br>High water loss (NRW) |
| | The proportion of customers connected to the sewer system (%) | 15.59 | 1 | Poor | Maybe difficult for sewer system construction due to the existence of pre-constructed infrastructures in most of the areas. |
| Akaki Kaliti | Proportion of piped water supply coverage (%) | 54 | 1 | Poor | Financial and other institutional problems<br>Problem with registration of number of costumers |
| | Water supply service duration (h day$^{-1}$) | 17.40 | 3 | Good | Power supply shortage (most of this area is supplied from groundwater) |
| | Per capita water consumption (L cap$^{-1}$ day$^{-1}$) | 39.40 | 2 | Fair | Urbanization is high in this area (construction boom)<br>Water loss (NRW) |
| | The proportion of customers connected to the sewer system (%) | 15.62 | 1 | Poor | Maybe difficult for sewer system construction due to the existence of pre-constructed infrastructures in most of the areas. |
| Gurdshola | Proportion of piped water supply coverage (%) | 100 | 5 | Excellent | The place near to the source (new ground water sources are constructed) and most of the areas are newly developed and construction of modern houses<br>Maybe the branch performance is good |
| | Water supply service duration (h day$^{-1}$) | 22.80 | 4 | Very good | Most of the area is supplied through gravity system |
| | Per capita water consumption (L cap$^{-1}$ day$^{-1}$) | 89.25 | 3 | Good | The coverage and service duration is good so that people will have access at least for more than 20 h and will have a chance to consume more water.<br>Water loss (NRW) |
| | The proportion of customers connected to the sewer system (%) | 6.94 | 1 | Poor | Maybe difficult for sewer system construction due to the existence of pre-constructed infrastructures in most of the areas. |

**Table 4.** *Cont.*

| Branch | Variable | Value | DWS Score | DWS Level | Major Challenges |
|---|---|---|---|---|---|
| Gulele | Proportion of piped water supply coverage (%) | 43 | 1 | Poor | Financial and other institutional problems<br>Problem with registration of number of customers |
| | Water supply service duration (h day$^{-1}$) | 13.90 | 2 | Fair | Topography (highly elevated area compared with other branches, with elevation variation of 2385–3015 meters AMSL)<br>Power supply shortage (to pump the water) |
| | Per capita water consumption (L cap$^{-1}$ day$^{-1}$) | 19.82 | 1 | Poor | Inadequate supply due to topography<br>Number of population is high in this area (more than 589,000).<br>Water loss (NRW) |
| | The proportion of customers connected to the sewer system (%) | 6.35 | 1 | Poor | Maybe difficult for sewer system construction due to the existence of pre-constructed infrastructure in most of the areas. |
| Megenagna | Proportion of piped water supply coverage (%) | 82 | 4 | Very good | Maybe the branch performance is good in terms of expansion of the coverage |
| | Water supply service duration (h day$^{-1}$) | 11.20 | 2 | Fair | Power supply shortage |
| | Per capita water consumption (L cap$^{-1}$ day$^{-1}$) | 64.33 | 3 | Good | High non-domestic consumption<br>Water loss (NRW) |
| | The proportion of customers connected to the sewer system (%) | 34.46 | 1 | Poor | This area has better sewer system than some other branches (though it is very small) due to it maybe being one of the older places in the city with a sewer system. |
| Mekanisa | Proportion of piped water supply coverage (%) | 78 | 3 | Good | Slum dwellers (some part of the area)<br>Financial and other institutional problems |
| | Water supply service duration (h day$^{-1}$) | 17.02 | 3 | Good | Power supply shortage (to pump the water) |
| | Per capita water consumption (L cap$^{-1}$ day$^{-1}$) | 50.67 | 3 | Good | Inadequate water supply due to power supply shortage<br>Water loss (NRW) |
| | The proportion of customers connected to the sewer system (%) | 23.35 | 1 | Poor | This area has a better sewer system than some other branches (though it is very small) due to it maybe being one of the older places in the city |
| Nifas Silk Lafto | Proportion of piped water supply coverage (%) | 99 | 5 | Excellent | Slum dwellers (some part of the area)<br>Household size might be high in this area due to dense population (number of customers small but many consumers) |
| | Water supply service duration (h day$^{-1}$) | 15.20 | 2 | Fair | Power supply shortage (to pump the water) |
| | Per capita water consumption (L cap$^{-1}$ day$^{-1}$) | 51.66 | 3 | Good | Inadequate water supply due to power supply shortage<br>Water loss (NRW) |
| | The proportion of customers connected to the sewer system (%) | 7.66 | 1 | Poor | Maybe difficult for sewer system construction due to the existence of pre constructed infrastructures in most of the areas. |

Another key issue pertains to the challenge of pre-existing infrastructure, and how it hinders effective network upgrades and expansion (i.e., the issue of "lock-in"). In many branches, this issue appears to be prohibiting easy construction, retrofitting, or maintenance of sewer and supply lines (Table 4), hindering the ability to efficiently expand the network, reduce NRW, and improve service coverage. Technological or infrastructural lock-in is the situation where the type and nature of existing structures or technologies make either retrofitting or replacement extremely time consuming or prohibitively expensive. In some cases, the only option is for complete system replacement, again at considerable cost. In older, well established districts, lock-in can be a major barrier to system enhancement or modification. This issue is not unique to this case, but may be exacerbated given institutional and financial constraints. Expansion in such areas could come at considerable expense, which the utility may not have access to. This issue may be less of an issue in new developments, where adaptable infrastructure can be built in from the start, but the worst performing areas are those that are already well developed and urbanized or slum areas. This could imply that improving service coverage in these areas, which experience the most acute problems, could be extremely difficult, costly, and time consuming.

A final major issue relates to the reliability of the power supply. In many (sub) branches, the issue of intermittent power supply is apparent. Loss of power impacts the continuity and quality of water services, thereby impacting domestic water security. Addressing the reliability of energy provision, like with the issues mentioned above, is a significant challenge in developing nations. Inadequate generating capacity, especially in the context of population growth, sub-optimal distribution infrastructure, and a relatively high proportion of people either unconnected or illegally connected add to this challenge. The issues surrounding institutional capacity and financing for development are relevant in this regard as well as for the water service sector as discussed above.

It is clear that there are a multitude of issues that can help explain the current domestic water security figures in Addis Ababa. Some issues are common between branches (and maybe between cities), and many branches experience multiple issues, exacerbating the problem and hindering improvement. However, through this work, it is shown which branches in Addis Ababa experience the most acute problems, and the nature of the main challenges on these branches is assessed (Table 4). Through this work, therefore, local city planners and the water utility have better information on where to target water service upgrades, both spatially and in which regard (e.g., power supply improvements or reducing NRW). While resources may not be immediately available for city-wide improvements, this research points towards those areas that could be prioritized, and which aspects should be considered as most urgent for improvement. Therefore, limited resources can be spent and directed more efficiently, leading to more effective water service improvements.

Regarding the framework itself, the aim was to develop a tool that could be broadly applied in diverse settings while still offering the potential for within-city levels of detail and giving meaningful output. There is therefore a fine balance between: (i) the development of relatively broad (generic) indicators that can be applied to other cities or urban and peri-urban areas without too much difficulty, regardless of the setting, increasing the wider applicability of this approach. While the indicators are broad, they can be tailored to specific settings through the use of relevant data and through narratives like those presented in Table 4, which add richness to the work. In this paper, the broad indicators are brought down to Addis city branch level using appropriate data. (ii) Developing much more specific indicators suited solely to a given case study. While this may have offered some additional insight on top of what is brought out here, it would preclude the easy transferability to other settings (e.g., outside eastern Africa, or even Ethiopia and Addis Ababa), which was a critical concern when developing the approach. It is believed that the presented framework achieves a good balance—being broad enough to apply in many settings, but in such a way that by using site-specific data, findings relevant for specific cases and potentially of operational value, especially when presented in a spatial manner and combined with a narrative such as Table 4, can be produced.

In a similar manner, the indicators and spatial reporting gives valuable information for operational water managers in Addis in order to assess which aspects of domestic water security fall short, and where. This will allow for more targeted intervention measures by water service providers, making for more targeted and efficient use of often scant resources in cities such as Addis. It can be argued that very specific indicators would add further richness to this overall message. However, the framework was to be developed to be as transferable as possible, meaning that a certain level of detail had to be "lost". While some issues described here may be cross cutting between cities (e.g., infrastructural lock-in, financial resource challenges), some may be very location specific (e.g., high-relief topography, power outages). A challenge related to some of these specific issues is how to quantify them in a meaningful manner (e.g., how to quantify infrastructural lock-in). Also, by developing very specific metrics (e.g., for topographic variability, power loss, etc.), the transferability of the framework could be compromised. One option would be to develop city-specific "add-ons" that would add local detail on top of that provided by this more general framework. This could generate the context specific insights desired. By developing more detailed indicators for this work, there was a risk of "pre-empting" issues (i.e., making a framework specifically aimed at a known case to give answers that were desired), and even worse, missing critical issues for other cities (i.e., limiting transferability). This could have the impact that other settings are reluctant to apply this framework.

The main potential drawback of this approach is the amount of data required to quantify the different indices (see Table 2). All these data may not be available in some locations, meaning that this method might have to be adapted to local circumstances and data availability. However, in principle, the framework is generally applicable and flexible depending on data availability. Another important consideration is the reliability of data that are obtained. Much of the data used here came from AAWSA, and it is assumed that the data are up to date and accurate, although there are no means to check this. Issues on data accessibility and reliability may also be issues elsewhere, and would need to be considered on a case-by-case basis. For this case, it was assumed that the data retrieved are accurate and reliable. Despite these issues, this framework allows for a relatively simple and rapid assessment of domestic water security, accounting for water supply, sanitation, and hygiene dimensions at a resolution greater than currently available. It is also a highly transferable method, and the results should be of wider interest to policy makers, planners, and water utilities.

## 6. Conclusions

Due to rapid urbanization, population growth, climate change, and the change in lifestyle because of economic growth, water security is a big challenge in cities, particularly in developing countries. Domestic water security assessment can help in giving a good assessment of the water supply, sanitation, and hygiene situation of a city by identifying major challenges that are location specific. This information can aid urban development and planning. However, quantifying domestic water security at the sub-city level is lacking. This study develops a novel domestic water security assessment framework to carry out such an assessment using an indicator based approach that assesses water supply, sanitation, and hygiene dimensions. The framework is applied to the city of Addis Ababa, Ethiopia. Results show considerable disparity in domestic water security within the city. The water supply dimension is better developed across the city; however, the sanitation dimension shows poor results generally, indicating an opportunity for development. Poorer and denser city branches report the worst results. Analysis highlights three main challenges faced throughout Addis in order to remedy the situation: (1) a lack of (institutional) capacity (financial, technical, and personnel) available to address the issues; (2) the impact of pre-existing infrastructure, leading to technological lock-in, and raising the cost, time, and effort for upgrades, maintenance, and retrofitting; and (3) an unreliable power supply, which has impacts on water supply and sanitation service reliability. Results for Addis Ababa, being spatially explicit, can help planners and policy makers target scant resources more effectively. The developed framework is also generally applicable, and can be applied to other urban and potentially peri-urban areas globally. It is suggested to carry out such assessments on a

regular basis (e.g., every 3–5 years) to judge the progress made and also across various urban centers in a country to compare the water security situation and develop an environment of competition among the utilities to improve the water supply, sanitation, and hygiene services for the people in their respective areas.

**Author Contributions:** Conceptualization, M.S.B. and V.R.S. Methodology, all authors. Validation, all authors. Formal Analysis, Y.T.A. Investigation, Y.T.A. Writing—original draft preparation, Y.T.A and J.S. Writing—review and editing, all authors. Visualization, Y.T.A. Supervision, M.S.B., V.R.S. and J.S.

**Funding:** This research was funded by the Netherlands Fellowship Programme (NFP).

**Acknowledgments:** We acknowledge Yidinekachew Shehmelo for his support in data collection and analysis.

**Conflicts of Interest:** The authors declare no conflict of interest.

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
