# Peer review of "Development of a Generic Domestic Water Security Index, and Its Application in Addis Ababa, Ethiopia"

_water, doi:10.3390/w11010037_

Round 1

Reviewer 1 Report

GENERAL COMMENTS

This paper is focused on the urban water security of the city of Addis Ababa.  The authors develop a suite quantitative of indicators/variables that can be used to assess the overall performance of the water supply system, the wastewater system, and the ability to achieve minimum hygiene levels.  Data is collected from the city water provider for a 5 year period, and values collected are normalized on a scale of 0-1 for the purpose of comparison. Results are presented for 10 sub-sections of the city, showing “good” performance for water supply, and “poor” performance for sanitation and hygiene.

This paper provides a closer look at water security issues in Addis Ababa, which is interesting.  However, the metrics selected are broadly defined (i.e., very similar to those that get applied at country or global scale analyses), which doesn’t lend much insight for a case study paper- even when the city is broken into sub-sections.  Fortunately, in their assessment of the results, the authors provide much unique and insightful commentary as to why the city scores as it does on water supply, sanitation and hygiene.  Given this is a case study paper, I would have expected the variables chosen to assess the city to be more detailed than that broad-level variables/indicators used here (see specific comments), and that a framework with more specific indicators/variables would then help make the tool developed here directly applicable to cities or urban water utilities who are interested in assessing and improving their water security.  As it stands, this paper comes across as city-level analysis using national-level indicators, which isn’t extremely exciting, and which is unfortunate since there is much richness that a city-level assessment framework could provide (as evidenced by the insights provided in Table 4).  While I understand the difficulties of data collection in developing world cities, as a single case study, I would have expected this extra effort from the authors. 

SPECIFIC COMMENTS

-          I appreciate the thought put into the dimensions, indicators and variables.  However, given that they are developed to cover the breadth of physical and human dimensions needed to ensure a “secure” water system in a city, I find it particularly problematic that the authors, even when focusing on a single case study, could not represent 1/2 of the variables listed as needed for characterization in Fig 1 for Sanitation, and could not represent 2/3 of the variables similarly listed for Hygiene.  If these are non-existent in Addis, would they be available in any other developing city (and if not, doesn’t this just make them “filler” variables that would never be used, but would theoretically be desirable)?  

-          Following on the above, if these variables are key to understanding water security, then I would have expected the authors to dig a little deeper than just the available data from a utility for some of these variables.  For instance, looking at WHO or subnational records of diarrhea in Addis, or trying to access or find data on education levels in the city from census data would have been more desirable than going to the effort of listing these as important for quantifying water security, and then dismissing them?

-          And lastly, why use a generalized “affordable tariff” value when working with a single city?  Surely this could be assessed for a single city?  The authors even mentioned that this could be assessed with a survey (Line 68)- was this done?  If so, this should be discussed.

-          In lines 203-207, the authors talk about the extremely low per capita consumption rates in several parts of the city, and is half of what the utility was projecting.  This either means that people are living in terrible conditions without access to even minimum volumes of water, or that they’re getting their water from somewhere else.  As presented, this framework appears to be focused on the water security of the city inhabitants (v.s. an assessment of the utility’s ability to meet demands).  As such, I’d expect more discussion and/or research to determine whether conditions really are this terrible, or if inhabitants are actually getting enough water, but through means not recorded by the utility (e.g., illegal connections or buying from water vendors). This seems important to sort out given the way the framework is presented.

-          In Section 4.1.5, the authors note that they obtained data “at each month” from AAWSA on water quality parameters.  At what frequency was water quality data collected by the utility?  Do they just test on one day each month?  Or was it daily data?  Or the mean of daily data over a month?  If it’s just one measurement per month, or the mean of daily data- this might be hiding violations that might be occurring at smaller time steps (but that could still be very important to city inhabitant health).

-          In Section 4.3.1, is the water available for basic hygiene assumed to be some specific fraction of the water consumption number?  And if so, does it then reduce the volume available for other uses (thus maybe double-counting across these two variables measured)?

-          I found the Challenges in Table 4 fascinating.  These highlight a number of conditions that prevent an urban area from achieving high water security (e.g., metering, land tenure issues, financial capacity, topography, etc.), but these are not explicitly represented as variables to be measured.  While it’s nice to have some very broadly defined variables so that it can be easily applied in other cities, I’d argue that the actual interesting insights annotated in Table 4 could also be broadly applicable across cities, and arguably provide much greater insight into why a city is less water secure than it would be, ideally, and would make this entire framework more directly useful to a water utility trying to improve water security for its customers.

TECHNICAL COMMENTS

-          This paper starts off well-written, but grammar issues and typos become more frequent as the manuscript develops (Section 2.3 and beyond).  A close re-read/revision would help bring this paper up to the expectations readers usually have for peer-reviewed journals.  

-          Also watch to make sure the correct meaning is being conveyed- for example, the first sentence of Section 2.4.2 states “About 80% of the water supplied to the customers is assumed to be wastewater.”  This literally says that the water being given to customers is 80% wastewater and 20% clean water?!

-          The last two columns in Table 1 seem redundant- the authors should either just pick one to include, or provide information in each such that they are not as directly repetitive of each other.

-          Why not put the expected variable units in Fig 1?  This would be a nice reference for anyone hoping to apply this methodology elsewhere.

-          In Section 2.3, there is “per capita” and “per-capita”.  The authors should stick to one style for the entire document (I’d suggest the former, since it’s much more common in the literature).

-          Equations 1 and 2 are awkward here- perhaps because the hyphen between “per” and “capita” at first comes across as a mathematical operator.  I’d argue these should not state their units on the left side of the equation- either put them in Fig 1, or include them in the narrative.  It would make these easier to read.

-          Its fairly common practice to state the relative direction of a numerical scale (in Table 2) when first mentioning the range of the scale.  A simple 1= poor to 5 = excellent, would suffice, and wouldn’t be redundant with Table 3.

-          The legends and figure text in Figures 2, 3 and 10 are very small.  These should be readable without needing to zoom in significantly.

-          I like Figure 9.  It’s a nice way to compare many sub-sections of the city against many indicators in an understandable way.

-          Finally, if possible, it would be interesting to see a map of population density, or age of building structures, or developed parcels vs. slums.  It would be helpful for giving the reader a sense of how development is unfolding across the city area.

-          The sections for Author Contributions, Funding, and Conflicts of Interest are not filled out here!

Author Response

We have responded to all the comments provided by reviewer 1 which has improved the quality of the article.

Reviewer 2 Report

Formal errors:

1.       The page and lines numbering is disrupted after the chapter 2.2

2.       Line 5, (Ch. 2.3.1) –water availability should be expressed in volume units per person and per time period, e.g. 1000 m3  c-1 year-1 (see also eq. 1)

3.       Do not use units in equations (eq. 1-4), just symbols like in eq. 5

4.       Explain symbols below the equation and state the units

5.       In Fig. 4, there are two viewpoints – a) the value of the indicator and b) the time development. Pls do not mix these two aspects, the graphs become disarranged. If you  want to demonstrate the timely development, insert a separate graph with a time-scale on the x-axis. In my opinion, you present the results (indicator values) for a specific date, thus the time development is redundant. This applies also for other graphs.

6.       I did not understand the “pre-existing infrastructure” issue – how it is prohibiting easy construction? It is because of the space arrangement, technology used, insufficient infrastructure capacity or other issues? Pls. Explain this more detailed in the paper.

Logical errors, comments, recommendations:

1.       The results of this (valid) study are not surprising at all, did the authors expect different results? Proposed methodology should be applied to more than one city, to show the ability of the methodology to be applied on various urban conditions and to differentiate various water supply/sewerage/sanitation development levels. OK, there are different administrative regions in the city of Addis Ababa analysed, but it is still the same city, same conditions.

2.       My opinion is, that the framework of the proposed assessment framework is far away from to being called “generic” (line 315). In different conditions a lot of indicator has to be modified and new additional indicators should be applied.

3.       The term “water security” in urban conditions is much more comprehensive, as proposed in paper – it comprises also crisis management (emergency plans), counter – terrorism (vandalism, sabotage…) measures, warning systems, much more detailed monitoring of water quality, ecosystems preservation (sustainability – groundwater extraction, surface water ecosystems…), also protection against water-related disasters (e.g. urban floods caused by storm rainfalls) etc.

4.       I miss in the paper some important information e.g. what is the type of the sewer system – combined or separate? How is the stormwater treated? Is the wastewater transported mainly by gravity or there is a large number or pumping stations? What is the accumulation (storage) volume of the drinking water supply network incl. reservoirs (water towers)? Is the delivered water declared as a drinking water or just utility water? The parameter range defined in lines 30-31 (page 6 in page 6 of the pdf document) is not sufficient for drinking water – what about nitrates, pesticides, heavy metals etc.?

5.       The benchmark of 0,40 USD is maybe suitable for this study, but generally I would rather recommend relative benchmark, based on the percentage of the average household income (eventually consumption basket).

Author Response

we have responded to the valuable comments provided which has really improved the quality of the paper.

Round 2

Reviewer 1 Report

My original comments were adequately addressed. I have no more suggestions prior to publication.

Reviewer 2 Report

Thank you for the explanations provided.

I think, the manuscript has been sufficiently improved following the indications and suggestions of the Reviewers.